# Assessment of the Nutritional Benefits and Aflatoxin B1 Adsorption Properties of Blackberry Seed Cold-Pressed Oil By-Product

**DOI:** 10.3390/foods13193140

**Published:** 2024-09-30

**Authors:** Jelena Miljanić, Saša Krstović, Lidija Perović, Jovana Kojić, Vanja Travičić, Branimir Bajac

**Affiliations:** 1Institute of Food Technology, University of Novi Sad, Bulevar Cara Lazara 1, 21000 Novi Sad, Serbia; lidija.perovic@fins.uns.ac.rs (L.P.); jovana.kojic@fins.uns.ac.rs (J.K.); 2Faculty of Agriculture, University of Novi Sad, Trg Dositeja Obradovića 8, 21000 Novi Sad, Serbia; sasa.krstovic@stocarstvo.edu.rs; 3Faculty of Technology Novi Sad, University of Novi Sad, Bulevar Cara Lazara 1, 21000 Novi Sad, Serbia; vanjaseregelj@tf.uns.ac.rs; 4BioSense Institute, University of Novi Sad, Dr Zorana Đinđića 1, 21000 Novi Sad, Serbia; branimir.bajac@biosense.rs

**Keywords:** blackberry seed cold-pressed oil cake, nutritional quality, mineral composition, aflatoxin B1-adsorbing properties, biosorbents characterization

## Abstract

This study explores the potential valorization of blackberry seed oil cake (BBSOC), a by-product of cold-pressed blackberry seed oil (*Rubus fruticosus* L.), as a nutritionally valuable material with aflatoxin B1 (AFB1) adsorption properties. The chemical and mineral composition, polyphenols, and antioxidant activity of BBSOC flour were assessed. BBSOC was found to be a significant source of fiber (62.09% dry weight) and essential minerals such as Fe (123.48 mg/kg), Mg (1281.40 mg/kg), K (3087.61 mg/kg), and Ca (1568.41 mg/kg). The high polyphenol content, especially ellagic acid, highlighted its biologically active potential. Moreover, BBSOC demonstrated effective biosorption of AFB1 under in vitro conditions at 37 °C, with adsorption efficiencies of 85.36% and 87.01% at pH 3 and 7, respectively. Characterization techniques including SEM, FTIR analysis, Boehm titration, and pH zero charge determination confirmed its AFB1 adsorbing properties. This valorization process reintroduces a secondary product into the food chain, supporting the circular economy and zero-waste concepts. Thus, BBSOC is nutritionally rich and effective in AFB1 biosorption, presenting potential applications as a food or feed additive.

## 1. Introduction

The global rise in industrial development and the rapid expansion of food industries result in the generation of significant amounts of waste each year. Valorization and reutilization of waste streams from food and agricultural products may offer a possible environmental and economic benefit. During the processing of berry fruits into different food products, high amounts of waste and side-off products are generated, commonly denoted as a pomace. The pomace consists of the remaining peel, seeds, and stalks and represents up to 20% of the fresh fruit weight used in the berry-making process. The seeds can be separated from the pomace while seed oil, as a product with added value, can be obtained by cold-pressing, leaving a valuable side stream [1]. Both product streams have attracted increasing attention in recent years, representing a small, but expanding segment of the market. Oilseed cakes produced through cold pressing are free of harmful solvents and rich in nutrients (proteins, dietary fibers, minerals, vitamins, antioxidants, etc.) [2]. The obtained pressed cake still contains bioactive compounds that can be potentially used in various food/feed applications generating minimum residual waste, aligning with zero waste and circular economy concepts. The utilization of defatted cakes as an animal feed supplement is well documented and confirmed in practice [3]. Due to the high contents of dietary fiber, proteins, and phenolic and other bioactive compounds, oil cake residual can be used for the valorization of functional ingredients or bioactive phytochemicals. Because of their high levels of dietary fiber, proteins, and phenolic and other bioactive compounds, oil cake residues can be utilized to enhance functional or bioactive ingredients. Recently, the application of oilseed cake in the form of dried, milled flour into novel functional foods such as bakery products [4], cookies [5], or specialty breads [6,7] has been investigated.

Over the last few years, there has been increasing interest in the health-promoting properties of blackberry (*Rubus fruticosus* L.) fruits. Notably, not only the fruit itself but also the seeds are recognized as rich sources of various bioactive compounds. Due to their nutritional value and potentially positive effects on health, there is growing attention toward the use of blackberry oilseed cakes [8]. Pressing residues of berry seeds have demonstrated high levels of antioxidant capacity, sometimes comparable to that found in herbs [9]. The study of Kosmala et al. indicated that blackberry press cake represents a significant source of dietary fiber and polyphenols, primarily ellagitannins with mono-, di-, and trimeric structures [10]. A feeding trial on rats revealed that extracted blackberry fiber exhibited the most significant health-beneficial properties among the examined preparations. This was evidenced by increased production of propionate and butyrate in the cecum, along with improved lipid levels in the blood [10].

Nowadays, a lot of attention has been paid to the application of biosorbents to mitigate different contaminants. The concept of utilizing cheap and easily available materials is becoming more popular. It involves the utilization of solid biowastes, mostly the by-products from the food industry and agricultural wastes [11]. Mycotoxin contamination of food and feed is a long-term global issue that can lead to increased human/animal exposure to adverse effects of these toxins. Mycotoxins are harmful secondary metabolites produced by fungi on agricultural commodities, either in the field or during storage. These toxins can lead to significant health issues, with some being recognized as carcinogens, prompting strict regulations worldwide. The predominant mycotoxins occurring in food/feed are aflatoxins, particularly aflatoxin B1 (AFB1) (classified as a Group 1 carcinogen), ochratoxin A, zearalenone, deoxynivalenol, T-2 toxin, and fumonisins. A variety of detoxification strategies have been developed over the years, considering the risks posed to animals and humans and the economic losses resulting from aflatoxin contamination in livestock. These procedures can be broadly categorized into chemical, physical, and biological methods [12,13,14]. The growing awareness of environmental protection and food safety drives the adoption of eco-friendly and safe approaches for eliminating mycotoxins in food production. In vitro aflatoxin adsorption by microorganisms such as yeast and lactic acid bacteria was reported with high efficiency and properties that depend on both the strain and the dosage [15]. Recent studies evaluated the efficacy of agri-food by-products as a low-cost biosorbent for mycotoxin decontamination [16,17]. The mentioned study evidenced that the best multi-mycotoxin adsorbent products were those rich in insoluble fibers (lignin, cellulose) and flavonoids, such as leaves and branches of olive trees, grape pomace (pulp and skin), artichoke, and almond by-products [16]. Avantaggiato et al. showed that a grape pomace (pulp and skin) from a red grape variety had the ability to rapidly and simultaneously remove various mycotoxins (AFB1, ZEA, OTA, and FB1) [18]. Numerous studies have been published on the efficacy of dietary fibers to prevent adverse effects of mycotoxins in animals [19,20,21]. Grape seed meal restored the inflammatory markers and performance in animals treated with AFB1, suggesting that grape waste is a promising feed source for mitigating the harmful effect of AFB1 [22]. Popescu et al. showed that the inclusion of grapeseed and sea buckthorn meal waste in pig diet, as by-products with antioxidant action, affected AFB1 and ochratoxin A biotransformation, leading to the enhancement of the antioxidant defense in the liver and kidneys [23].

Oil cakes, obtained after oil extraction from different plant materials such as mustard, neem, and pongamia, have previously been utilized for the removal of heavy metals, namely nickel [24], zinc [25], copper, and cadmium [26], from aqueous solutions. Within the authors’ disposal, there is no literature data on the in vitro ability of BBSOC to bind different mycotoxins. Therefore, the main goal of this research was to test BBSOC, as a valuable nutritional source, for their capacity to adsorb AFB1 at pH 3 and 7 expected in the digestive tract of monogastric animals and humans. Different sorbent dosages were selected in order to investigate the effect of the adsorbent on the capacity of BBSOC. Furthermore, the chemical composition, nutritional value, and antioxidant activity of BBSOC were studied to valorize those waste streams through value-added utilization as beneficial food/feed ingredients.

## 2. Materials and Methods

### 2.1. Material

BBSOC was obtained from a local cold-pressed oil producer in Novi Sad, Serbia. Samples were defatted by rinsing twice in hot water. Afterward, the material was dried for 24 h at 60 °C and sieved to the desired particle size (<500 μm).

### 2.2. Chemical Characterization of BBSOC

Moisture content, ash, total dietary fibers, protein, fat, and total sugar content were determined according to AOAC official methods [27]. Acid detergent lignin (ADL), acid detergent fiber (ADF), and neutral detergent fiber (NDF) content were determined by the Van Soest et al. (1991) method, using the ANKOM 2000 Fiber Analyzer (ANKOM Technology, Macedon, NY, USA) [28]. Cellulose and hemicellulose content were calculated using ADF, NDF, and ADL results.

#### Minerals Content

The mineral content, including potassium (K), calcium (Ca), magnesium (Mg), iron (Fe), and zinc (Zn), in the BBSOC was analyzed using an atomic absorption spectrometer (Varian Spectra AA 10, Varian Techtron Pty Limited, Melbourne, Australia). The metal concentrations were determined in accordance with a standard procedure [29]. Each measurement was performed in triplicate, and the results are expressed as average values.

### 2.3. Phenolic Compounds

Polyphenol extraction followed the method outlined in Kosmala et al. [10], conducted in three stages using a 70/30 (*v*/*v*) acetone/water solution. Initially, 0.5 g of ground BBSOC was mixed with 4 mL of the solvent and sonicated for 15 min. After centrifugation at 4800× *g*, the supernatant was collected in a flask. The procedure was repeated twice more, using 3 mL of the solvent each time.

#### HPLC-DAD

The HPLC method, adapted from Tumbas Šaponjac et al. [30] with slight modifications, was employed to quantify individual polyphenols. The analysis was conducted using a liquid chromatograph (Agilent 1290, Agilent, Santa Clara, CA, USA) coupled with a DAD detector and a Zorbax Eclipse XDB-C18 column, 1.8 μm, 4.6 × 250 mm (Agilent, Santa Clara, CA, USA). The injection volume was 5 μL. Polyphenol separation was achieved using methanol (A) and 0.5% (*v*/*v*) formic acid in water (B) as mobile phases, with a flow rate of 0.3 mL/min. The solvent gradient program was as follows: initial, 3% A; 0–13 min, 3% A; 13–25 min, 5% A; 25–34 min, 6% A; 34–35 min, 9% A; 35–52 min, 10% A; 52–75 min, 25% A; 75–95 min, 45% A; 95–110 min, 60% A; 110–130 min, 90% A; and 130–133 min, 3% A. Prior to injection, the samples were diluted in a mobile phase mixture (10A:90B %, *v*/*v*) and filtered through a 0.22 μm RC syringe filter. The wavelengths for detection were set at 280 nm, 320 nm, and 362 nm. The results are expressed in µg of compound per gram of seed (µg/g). All experiments were performed in triplicate.

### 2.4. Determination of Antioxidant Activity

The antioxidant activity was assessed following three different assays: 2,2-diphenyl-1-picrylhydrazyl (DPPH), reducing power (RP), and 2,2′-azino-bis-3-ethylbenzothiazoline-6-sulphonic acid (ABTS) as described by Tumbas Šaponjac et al. [30]. The tests were performed on 70% methanol extracts of BBSOC. The antioxidant activity was expressed as millimoles of Trolox equivalent (TE) per 100g of BBSOC. All the analyses were performed in triplicate. 

#### 2.4.1. DPPH Radical Scavenging Assay

The DPPH radical scavenging assay was conducted using a spectrophotometric method. In summary, 250 μL of a 0.89 mM DPPH^•^ solution in methanol was combined with 10 μL of the sample in each well of a microplate. After incubating the mixture in the dark at room temperature for 50 min, the absorbance was recorded at 515 nm. Methanol was used as the blank. 

DPPH radical scavenging activity values were calculated using the following equation: DPPH = [(A_control_ − A_sample_)/A_control_] × 100
where A_control_ represents the absorbance of the blank and A_sample_ denotes the absorbance of the sample.

#### 2.4.2. Reducing Power

Reducing power (RP) was determined by a spectrophotometric method adapted for a 96-well microplate [31]. In this procedure, 25 μL of the sample or 25 μL of water (for the blank test), 25 μL of sodium phosphate buffer (pH 6.6), and 25 μL of 1% potassium ferricyanide solution were combined and incubated in a water bath at 50 °C for 20 min. After incubation, the mixture was cooled, and 25 μL of 10% trichloroacetic acid was added. The solutions were then centrifuged at 2470× *g* for 10 min. Following centrifugation, 50 μL of the supernatant was mixed with 50 μL of distilled water and 10 μL of 0.1% ferric chloride solution in a microplate. Absorbance readings were taken immediately at 700 nm.

#### 2.4.3. ABTS Radical Scavenging Assay

The ABTS radical scavenging assay was conducted as follows [32]. The absorbances of 250 μL activated ABTS^+•^ (with MnO_2_) before and 35 min (incubated at 25 °C) after the addition of 2 μL of the sample were measured at 414 nm. Distilled water was used as the blank. 

### 2.5. Biosorbents Characterization of BBSOC

#### 2.5.1. SEM and FTIR Analysis

The surface morphology of the BBSOC was examined using scanning electron microscopy (SEM) (Tabletop microscope TM3030, Hitachi, Tokyo, Japan). A SEM image analysis was performed before and after adsorption. A Fourier transform infrared spectroscopy with attenuated total reflection technique (FTIR-ATR) (Alpha Bruker Optics instrument, Bruker Optics, Ettlingen, Germany) was used to determine the chemical functional groups on the oil cake surfaces. Analysis was conducted in the spectral range from 400 to 4000 cm^−1^.

#### 2.5.2. Boehm Titration and pH of Zero Charge

The acidic and basic groups on the surface of BBSOC were determined through acid-base titration experiments, as described by Boehm [33]. In these experiments, 1 g of the sample was added to 50 mL of different solutions, including sodium hydroxide, sodium carbonate, sodium bicarbonate, and hydrochloric acid. The mixtures were sealed and agitated for 24 h, after which they were filtered. Five milliliters of each filtrate were then pipetted, and the excess acid or base was titrated with either hydrochloric acid (HCl) or sodium hydroxide (NaOH), respectively. The determination of acidic sites was based on the subsequent premises: NaOH neutralizes carboxylic, lactonic, and phenolic groups; Na_2_CO_3_ neutralizes carboxylic and lactonic groups; and NaHCO_3_ neutralizes only carboxylic groups. The basic sites were quantified based on the amount of hydrochloric acid that reacted with the carbon.

The point of zero charge (pH_pzc_) was determined following the method described by Khan et al. [24]. One gram of BBSOC was suspended in a 10^−3^ mol/dm^3^ KNO_3_ solution for 24 h. This suspension was then divided into eight 60 mL portions; each adjusted to different pH levels (2.23, 3.14, 4.02, 5.12, 6.2, 7.13, 8.02, and 9.11). Each portion was further split into four equal parts. To two of these sets, 0.3 g of KNO_3_ was added, while the other two sets received no additional nitrate. All samples were left for 6 h. Following this period, the pH of each reference and test suspension was measured and recorded as the initial and final pH, respectively. The pairs without added nitrate served as the reference samples, while those with added nitrate were the test samples. The pH_pzc_ was determined by plotting ΔpH (pH_f_ − pH_i_) against the initial pH and identifying the pH value where ΔpH equals zero.

### 2.6. AFB1 Adsorption Experiments

BBSOC samples were weighed into 20 mL silanized amber glass vials and suspended in a mycotoxin working solution (AFB1 concentration = 2 mg/mL), buffered at either pH 3 or pH 7. The concentration of the solid phase (BBSOC) ranged from 2 to 35 mg/mL (2, 6, 12, 18, 25, 30, and 35 mg/mL). The suspensions were vortexed for a few seconds and then shaken for 120 min in a shaker (KS 4000, IKA^®^-Werke GmbH & Co. KG, Staufen, Germany) at 37 °C and 250 rpm. After incubation, 1 mL of each suspension was transferred into an Eppendorf tube and centrifuged for 20 min (18,000× *g* and 25 °C). 

The AFB1 working solution in the buffer without the BBSOC was used as a blank control. This control followed the same procedure to assess AFB1 stability in the buffer and to evaluate any nonspecific adsorption. No chemical precipitation or nonspecific adsorption of AFB1 was detected during the experiment.

#### AFB1 HPLC Determination

The supernatants were evaporated and reconstituted in the mobile phase for the determination of residual mycotoxin content using HPLC with fluorescence detection. A 20 µL sample or standard was injected, and aflatoxins were separated using a Hypersil ODS C-18 column (150 × 4.6 mm i.d., particle size 5 μm) with a mobile phase of water (25:75, *v*/*v*) at a flow rate of 1.0 mL/min. The column temperature was set at 40 °C. Fluorescence detection was performed at an excitation wavelength of 365 nm and an emission wavelength of 445 nm. The total run time was 10 min. Method validation included the analysis of blank samples, spiked samples, and control standards in each run. In-house validation was also conducted, using certified reference material TR-A100 (Trilogy, Newburyport, Massachusetts, USA), with a certified value of 19.0 ± 2.3 µg/kg. The method showed an average trueness of 105.6%. The limit of quantification (LOQ) for AFB1 was calculated based on the signal-to-noise ratio and was found to be 2.5 ng/mL. This LOQ was experimentally verified by spiking a blank sample with a mycotoxin standard. The method’s linearity was confirmed by constructing a calibration curve with a regression coefficient of R^2^ > 0.999. All analyses were carried out in duplicate.

The adsorption efficiency is represented by an adsorption index, where C_i_ indicates the initial concentration and C_eq_ is the equilibrium concentration of AFB1: Adsorption index (%) = [(C_i_ − C_eq_)/C_i_] × 100

### 2.7. Statistical Analysis

The data were analyzed using a two-way analysis of variance (ANOVA) and Duncan’s multiple comparison tests (*p*-values < 0.05 were considered significant). All statistical analyses were conducted with STATISTICA 13.0 (StatSoft, Palo Alto, CA, USA). 

## 3. Results and Discussion

### 3.1. Chemical and Mineral Components

The results of the chemical analysis of BBSOC are presented in Table 1. To assess the nutritional profile of BBSOC the recommended daily intake of nutrients was added to the table. The chemical composition of oil cake can vary significantly, primarily influenced by the characteristics of the raw material (such as the vegetable species, plant nutrition, and climatic conditions). Variations in the pretreatment technology of oilseeds, such as impurity removal and flaking, also contribute substantially to this variability. The type of technology used for oil extraction (mechanical pressing or solvent-based extraction) and the efficiency of oil recovery from seeds are directly related to the chemical composition of the oil cake. A higher yield of oil results in a higher concentration of other components in the cake. The flour obtained after pressing blackberry seeds is characterized by its high fiber content (62.09%) and relatively low levels of carbohydrates, fats, and proteins (7.36%, 6.15%, and 13.20%, respectively) [34]. The levels of protein and ash in blackberry seeds were similar in the study by Kosmala et al. [10]. In the mentioned study, the fiber content in blackberry fiber, extracted blackberry fiber, and defatted blackberry seeds was 60.3%, 65.5%, and 72.2%, respectively.

Depending on the source and intake of dietary fiber, published research has demonstrated several health benefits, including reducing blood sugar and cholesterol levels, lowering the risk of cardiovascular diseases, aiding in weight loss, improving insulin regulation, suppressing type 2 diabetes, and generally enhancing immunity [36]. The recommended daily fiber intake for adults is approximately 25 to 30 g, derived from food rather than supplements. To meet this requirement, adults could consume a product containing about 50 g of BBSOC flour, which provides an average of 31.04 g of dietary fiber. Additionally, BBSOC flour after pressing is rich in insoluble fibers (55.85%), including cellulose, hemicellulose, and lignin. During food digestion, insoluble fibers bind water in chyme, softening it and facilitating easier passage with less strain. They can also improve overall intestinal health and motility, positively affect insulin levels, and help reduce the risk of diabetes [37]. According to the National Research Council, the appropriate proportion of ADF in ruminant nutrition is crucial, as inadequate levels can cause a decrease in milk fat content, acidosis, and rennet displacement [38]. NDF is also important, as it is correlated highly with rumination and overall feed consumption. Increasing the proportion of NDF in a cow’s diet can decrease feed intake. Lignin content significantly affects the digestibility of feed [39]. The chemical composition of BBSOC shows they contain 55.85% NDF. However, the high content of ADF (50.89%) and especially lignin (24.75%) suggests that using high BBSOC ratios in ruminant diets could decrease food consumption and meal digestibility. Conversely, if used as a dietary supplement, for example, to adsorb mycotoxins, it would not negatively impact food consumption or meal digestibility.

Macro and micronutrients are essential for the growth, proper functioning, and health of both humans and animals. Adequate mineral intake is crucial for maintaining health and preventing or treating cardiovascular diseases, such as arterial hypertension and bone demineralization. Table 2. presents the mineral content of the BBSOC, highlighting the dominant presence of iron and magnesium. The iron content in BBSOC flour was 123.48 mg/kg, indicating that consuming BBSOC can contribute to meeting part of the recommended daily intake of 14 mg. Krstić et al. revealed that among the examined fruit seeds, blackberry seeds contained the highest iron content at 97 mg/kg dry weight (d.w.) [40]. Iron is an essential mineral found in every cell of living organisms and is necessary for the formation of hemoglobin, a protein that supplies oxygen to the blood. Iron deficiency, particularly in women and children, can lead to anemia [41]. Given its high iron content, BBSOC flour is suitable for individuals with iron deficiency anemia. Magnesium is crucial for cellular production and growth and is involved in hundreds of enzymatic processes in all plants and animals, including humans. Magnesium helps produce ATP energy in cells and activates protein production, including the synthesis of DNA structures. Its roles include energy transport, protein synthesis, nerve impulse transmission, muscle relaxation, and participation in numerous chemical reactions, including body temperature regulation. The recommended daily intake of magnesium is 375 mg while BBSOC (100g) contains 128.14 mg. The calcium content in BBSOC flour is 1568.41 mg/kg, while the recommended daily intake is 800 mg. Calcium is one of the more abundant minerals in berry seeds and is essential for maintaining optimal blood pressure, normal brain function, and preventing blood clotting [40]. Potassium is one of the seven essential macrominerals, with the human body requiring a minimum of 100 mg daily to maintain critical functions. A higher intake of potassium has been linked to a 20% decrease in the risk of overall mortality. It also contributes to lowering the risk of stroke, reducing blood pressure, preventing muscle mass loss, preserving bone mineral density, and decreasing kidney stone formation. Potassium’s key roles include regulating fluid balance and managing the electrical activity of the heart and muscles. The recommended daily intake for potassium is 2000 mg, although the World Health Organization advises an intake of 3510 mg per day for adults [42]. In the BBSOC flour sample, a significant potassium content of 3087.61 mg/kg was measured. As noted by Grzelak-Błaszczyk et al., defatted strawberry seeds also contain considerable amounts of calcium and potassium, their primary minerals, ranging between 4.1 and 6.0 g/kg d.w. and 3.1 to 4.8 g/kg d.w., respectively, depending on the season [43]. Moreover, these seeds have notable magnesium levels (1.9–2.3 g/kg d.w.) and iron (79–148.8 mg/kg d.w.), while sodium content remains low. The zinc content in BBSOC was found to be 14.51 mg/kg. Zinc deficiency, which affects around 155 million children under the age of five worldwide, is associated with stunted growth [42].

Therefore, innovative strategies are needed to ensure adequate levels of micronutrients in food and animal feed, one of which involves supplementing diets with mineral-rich by-products. The results indicate that BBSOC is an excellent source of both micro- and macronutrients.

### 3.2. Polyphenol Compounds and Antioxidant Activity

Phenolic compounds in current literature primarily emphasize berry fruits rather than their seeds. Berry seeds, typically incorporated within the fleshy mesocarp and consumed along with the fruit, significantly contribute to the total bioactive compound content of berries. They account for approximately 60–70% of the biological activity of these fruits [44]. In addition to fibers and minerals, BBSOC is rich in polyphenols, predominantly ellagitannins, ellagic acid derivatives, and ellagic acid itself (Table 2).

The analysis revealed that the BBSOC sample contained significant amounts of ellagic acid, quantified at 827.2 ± 21.2 µg/g. This finding closely aligns with the results reported by Choe et al. [45]., who highlighted that ellagitannins, derivatives of ellagic acid, and ellagic acid itself are the main polyphenols in cold-pressed BBSOC. Other phenolic compounds were detected only in trace amounts. In their study, the ellagic acid content was reported as 653.81 µg/g of the seed cake, slightly lower than our current findings. This overlap in data supports the robustness of these results, illustrating the key role of ellagic acid and its derivatives in the polyphenolic composition of blackberry seed oil cake across different studies. Hager et al. and Kosmala et al. also identified only ellagitannins, derivatives of ellagic acid, and ellagic acid in blackberry seeds, confirming that these phenolic compounds are the most significant in terms of content [10,46]. Ellagic acid, primarily found in blackberry seeds as ellagitannins, is a hydrolyzable tannin precursor that releases ellagic acid upon metabolism in the human gastrointestinal tract. While the acidic stomach environment results in minimal hydrolysis, significant release occurs in the small intestine, where ellagitannins are converted into ellagic acid and subsequently metabolized into bioactive urolithins [47]. These compounds exhibit neuroprotective effects, potentially benefiting conditions such as Parkinson’s disease by supporting dopamine levels through antioxidant-rich diets, such as blackberry seeds [48]. 

Additionally, the sample exhibited levels of gallic acid and protocatechuic acid at 37.2 ± 4.3 µg/g and 49.8 ± 5.2 µg/g, respectively. When compared with the data obtained from methanol–acetone–water extracts in the study of Ayoub [49], our findings for gallic acid and protocatechuic acid showed a close alignment. BBSOC demonstrated strong antioxidant activity and a significant presence of polyphenols such as quercetin and p-coumaric acid, consistent with results reported in similar studies. The use of methodologies such as HPLC-DAD for polyphenol profiling yielded comparable results, underscoring the reproducibility of these measurements across different research efforts [50]. Overall, these results emphasize the robust phenolic content of BBSOC and confirm its potential as a source of bioactive compounds.

The antioxidant activity was measured using ABTS (379.97 mmol TE/g), RP (50.34 mmol TE/g), and DPPH (63.86 mmol TE/g) assays (Figure 1). It is important to note that comparing these results with other scientific studies using the same assays is often impractical due to the lack of standardized methods for determining antioxidant activity. For instance, it is not feasible to directly compare the results from Park et al. [51], who stated that blackberry seed extract concentrations of 40.9 and 92.2 µg/mL are sufficient to inhibit 50% of ABTS and DPPH radicals, thus determining the antioxidant value of the extract. In contrast, Sariburun et al. reported that the antioxidant activity determined by the ABTS assay for whole blackberry fruit ranged from 64.36 to 146.89 µmol TE/g, while for the DPPH assay, it ranged from 64.14 to 177.11 µmol TE/g [52]. Also, the ABTS value obtained from the current study is compatible with the ABTS values of blackberry seed flour that Choe et al. previously reported (266.82 μmol TE/g) [45]. Sariburun et al. noted that the antioxidant value depends on the blackberry variety and extraction technique, but polyphenols are the main contributors to antioxidant activity [52].

The antioxidant properties of polyphenols and polyphenol-rich foods (such as blackberry seed oil cake) can play an important role in cancer prevention if such foods are consumed in sufficient quantities. The antiproliferative activity of ellagic acid and other polyphenols is often linked to their antioxidant potential and ability to scavenge reactive oxygen species (ROS) and reactive nitrogen species (RNS) through aromatic polyphenol rings. This mechanism helps prevent and/or reduce oxidative stress in cells, which can cause DNA damage in healthy cells. If not properly regulated, excessive oxidative stress can eventually lead to the formation of tumor cells [53]. 

### 3.3. Characterization of BBSOC as a Biosorbent for AFB1 Removal

The characterization of biosorbents BBSOC used for the removal of AFB1 from solutions was conducted using scanning electron microscopy (SEM) analysis, FTIR analysis, Boehm titration, and determination of the point of zero charge (pHpzc).

#### 3.3.1. SEM and FTIR Characterization

The SEM micrograph of the biosorbent before biosorption is shown in Figure 2a, as well as the SEM micrograph of the same biosorbent after AFB1 biosorption (Figure 2b).

Based on the SEM analysis, which examined the morphology of the tested samples, differences were observed in the structure of the biosorbents before and after AFB1 adsorption. The surface is partly layered, and partly embossed, with smaller and larger protrusions. These irregularities on the surface enhance the contact area. The surface structure after sorption is rougher and more uneven, probably due to the surface adsorption of AFB1. To determine whether there are changes at the level of functional groups, which occurred as a result of binding AFB1, FTIR analysis of the biosorbent was performed.

FTIR analysis is based on the absorption of specific frequencies characteristic of the molecular structure, which are recorded as peaks in the infrared spectrum of a compound. The identification of certain functional groups is achieved by analyzing the obtained wavenumber values. Food and agricultural by-products, as organic matter, primarily consist of cellulose and lignin and may contain functional groups such as carboxylic, phenolic, ether groups, alcohols, aldehydes, and ketones. BBSOC was characterized using FTIR to identify the functional groups that represent active binding sites for AFB1. The FTIR spectra obtained for BBSOC before and after AFB1 biosorption, in the range of 4000–500 cm^−1^, are presented in Figure 3.

The FTIR spectrum revealed a broad, intense band in the range of 3200 to 3600 cm⁻^1^, corresponding to the valence vibration of the O–H group, which originates from the alcohol, phenolic, and carboxyl functional groups present in cellulose and lignin [54]. The spectra before and after biosorption were very similar, with changes observed in the intensity of individual peaks (Figure 3). The most significant changes occurred in the 3500–3000 cm⁻^1^, 1600 cm⁻^1^, and 1050 cm⁻^1^ regions, indicating that carboxyl and hydroxyl groups play a role in biosorption. The peak at 2900 cm⁻^1^ corresponds to the vibrational stretching of the –CH bonds in lignin and hemicelluloses. Another important peak appeared in the 1600–1800 cm⁻^1^ range, related to carbonyl stretching vibrations in the carboxyl groups of pectin and hemicellulose. An intense peak at 1050 cm⁻^1^ is attributed to the C-O-C glycosidic bond in polysaccharides such as cellulose [55]. Based on the FTIR spectra, it can be concluded that various functional groups, primarily carboxyl and hydroxyl, are involved in the toxin-binding process.

#### 3.3.2. Boehm Titration and Zero Charge pH (pHpzc)

Surface functional groups are important in sorption processes. These groups are typically classified as acidic or basic, influencing the surface charge of the biosorbent and subsequently the efficiency of biosorption (Table 3).

Based on the results, it can be concluded that the surface of BBSOC primarily contains acidic groups, including carboxylic, phenolic, and lactone groups, with a smaller proportion of basic groups. This smaller share of basic functional groups, in comparison to acidic oxygen functional groups, aligns with the structural composition of the biosorbent, which is mainly composed of cellulose, hemicellulose, and lignin, as well as the absence of nitrogen or protein (Table 1). The surface charge of BBSOC is a function of pH value. It was determined that the pH value at the point of zero charge of the biosorbent is 4.9 (Figure 4). This value is consistent with the results of the Boehm titration, which indicates a predominance of acidic groups on the surfaces of the biosorbent.

The pH value at the point of zero charge (pHpzc) was determined based on the change in the pH of a KNO_3_ solution affected by the biosorbent. Figure 4. shows the dependence curve of the final pH (pH_f_) relative to the initial pH (pH_i_) for three different ionic strengths of the electrolyte. 

The curves at different KNO_3_ concentrations nearly overlapped, indicating that KNO_3_ was inert to the investigated biosorbent, as the ions of this electrolyte (K⁺ and NO₃⁻) were not specifically adsorbed on its surface, involving only physical rather than chemical sorption [56]. The plateau of the curves corresponds to the point of zero charge (pHpzc), which is 4.9. In the first segment of the curve, with pH values of 1–2, the biosorbent surface was positively charged. This is followed by a flattened plateau at pH values of 4–9, where the surface was neutral, meaning that the sum of the negative charge equals the sum of the positive charge. At higher pH values, the surface acquired a negative charge.

AFB1 is a highly polar molecule with a positively charged surface [57]. In the study by Vázquez-Durán et al., zeolite with a pHpzc close to 9 was found to be an excellent inorganic material for binding AFB1 [58]. The pH of the zeolite, ranging from 2 to 8.8, exhibited a consistent positive charge, indicating that AFB1 adsorption is not primarily driven by electrostatic attraction. It can be concluded that the interaction between AFB1 and the biosorbent is based on a combination of chemical and physical mechanisms, including physicochemical interactions and complexation, which result in the immobilization of most AFB1 molecules.

#### 3.3.3. In Vitro Adsorption of AFB1 Using BBSOC

The results of AFB1 adsorption using BBSOC are shown in Table 4.

By varying the dose of the biosorbent, it was observed that increasing the mass up to 30 mg enhanced the removal efficiency of AFB1 (Table 4), as the number of available biosorption sites increased with the biosorbent dose [59]. The optimum contact time for reaching equilibrium was 120 min, likely due to the saturation of active sites, which limited further adsorption. The adsorption efficiency was 85.36% at pH 3 and 87.01% at pH 7. However, increasing the biosorbent mass to 35 mg did not lead to a greater reduction in AFB1 levels. Similar AFB1 adsorption values (83%) were reported by Greco et al. using grape seeds as a biosorbent [16]. Additionally, the study by Vázquez-Durán et al. demonstrated that lettuce residues achieved the highest AFB1 removal efficiency (95%) at pH 7 [58].

The results indicate that the pH of the environment does not significantly impact the adsorption of AFB1. The high removal efficiency of the mycotoxin AFB1 (>85%) using unmodified natural material, BBSOC, suggests that it could serve as a viable alternative to many currently available but less effective materials on the market. While modifying the material with acids or bases could enhance its removal efficiency, such changes would compromise its natural status. This would make it unsuitable for use as a supplement in the production of clean-label products, which are increasingly sought after by consumers worldwide [60].

## 4. Conclusions

The advantage of BBSOC lies in its complex composition, which combines nutritional quality, antioxidant potential, and adsorbent properties. Novel food ingredients rich in natural antioxidants enhance their value-added utilization as beneficial components in food products. These ingredients can be applied with the primary aim of ensuring food and feed safety, as well as promoting human and animal health. The chemical and mineral composition of BBSOC flour reveals it to be a significant source of fiber and essential nutrients. The high polyphenol content, especially ellagic acid, underscores its biologically active potential and nutritional quality. The overall assessment of BBSOC’s adsorption performance demonstrates its high effectiveness as a novel, economical, and environmentally friendly biosorbent for the efficient removal of AFB1. Following in vitro studies, additional in vivo studies are required to demonstrate the effectiveness of reducing the toxic effects of mycotoxins while preserving micronutrient bioavailability. However, further studies are needed to better understand the complex mechanisms of AFB1 adsorption on various matrices beyond just aqueous mediums.

## Figures and Tables

**Figure 1 foods-13-03140-f001:**
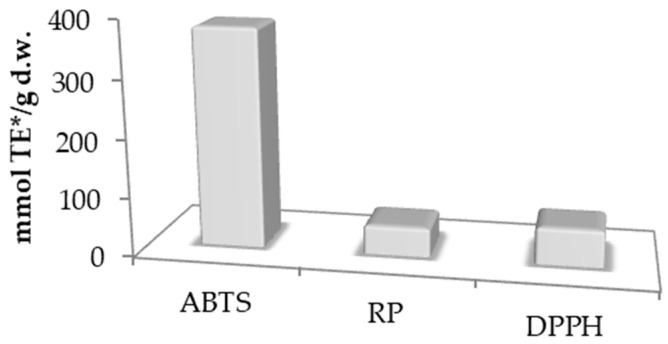
Antioxidative activity of BBSOC.

**Figure 2 foods-13-03140-f002:**
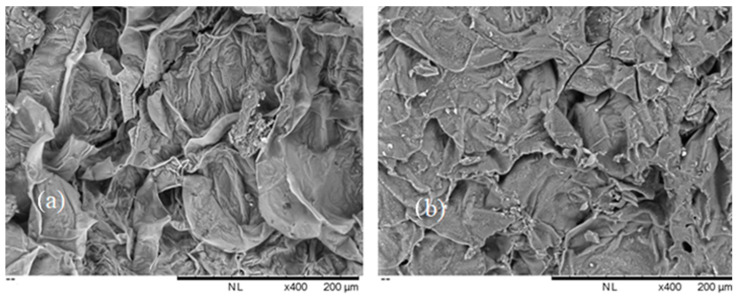
SEM micrograph of the BBSOC (**a**) before and (**b**) after AFB1 adsorption (×400).

**Figure 3 foods-13-03140-f003:**
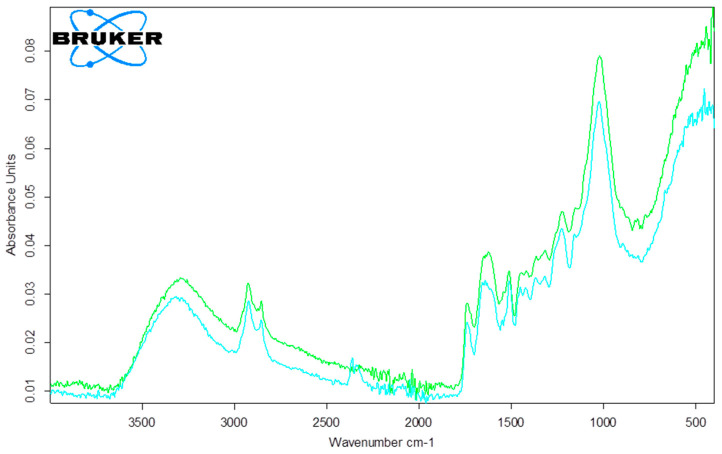
FTIR spectra of the BBSOC before (green line) and after AFB1 adsorption (blue line).

**Figure 4 foods-13-03140-f004:**
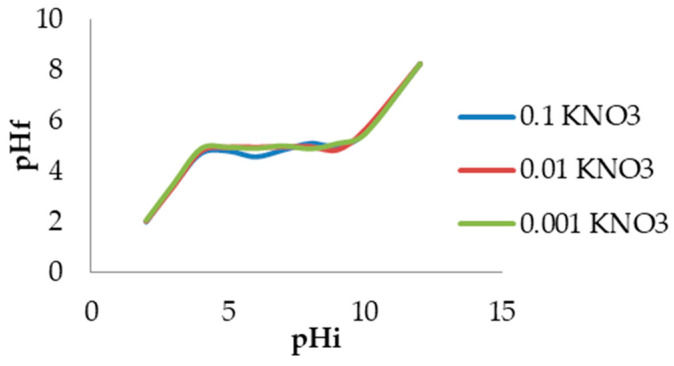
Change in pHf with change in pHi of BBSOC for different ionic strengths of KNO_3_ solution.

**Table 1 foods-13-03140-t001:** Chemical and mineral composition of BBSOC.

**Nutrients (g/100g)**	**Content**	****** Daily Reference Intake, g**
Moisture	3.98 ± 0.01	-
Ash	1.85 ± 0.08	-
Proteins	13.20 ± 1.02	50
Fats	6.15 ± 0.08	70
Carbohydrates	7.36 ± 0.09	260
Dietary fiber	62.09 ± 0.95	30
ADF *	50.89 ± 0.25	
ADL **	24.75 ± 0.28	
NDF ***	55.86 ± 0.27	
Cellulose	31.11 ± 0.01	
Hemicellulose	4.97 ± 0.01	
**Minerals (mg/kg)**	**Content**	****** Daily Reference Intake, mg**
Potassium (K)	3087.61 ± 15.99	2000
Calcium (Ca)	1568.41 ± 54.52	800
Magnesium (Mg)	1281.40 ± 118.02	375
Iron (Fe)	123.48 ± 8.12	14
Zinc (Zn)	14.51 ± 2.11	10

*ADF—Acid Detergent Fiber, ** ADL—Acid Detergent Lignin, and *** NDF—Neutral Detergent Fiber, **** Regulation (EU) No. 1169/2011 [35].

**Table 2 foods-13-03140-t002:** Phenolic compound contents of BBSOC.

Phenolic Compound	µg/g
Gallic acid	37.2 ± 4.3
Protocatechuic acid	49.8 ± 5.2
p-coumaric acid	4.04 ± 0.7
Ellagic acid	827.2 ± 21.2
Quercetin	20.2 ± 4.8
Qu-3 rhamnosid	26.2 ± 5.1

**Table 3 foods-13-03140-t003:** Acidic and basic group content on the biosorbent surface.

Biosorbent	Functional Groups				
	Acidic Groups (mmol/g)				Base Groups (mmol/g)
	Phenolic	Lactone	Carboxylic	Total	
BBSOC	0.10 ± 0.02	0.85 ± 0.1	0.63 ± 0.07	1.58 ± 0.12	1.18 ± 0.11

**Table 4 foods-13-03140-t004:** Adsorption index of AFB1 by BBSOC at pH 3 and 7.

AFB1	Adsorption Index %	
mg/mL	pH 3	pH 7
2	3.54 ± 0.34 ^a^	4.23 ± 0,.21 ^a^
6	7.50 ± 0.82 ^b^	11.62 ± 0.91 ^b^
12	24.36 ± 2.67 ^c^	29.35 ± 2.31 ^c^
18	48.35 ± 3.85 ^d^	50.56 ± 4.69 ^d^
25	74.42 ± 5.36 ^e^	79.24 ± 5.28 ^e^
30	85.36 ± 7.11 ^f^	87.01 ± 8.15 ^f^
35	84.91 ± 8.02 ^f^	86.87 ± 8.94 ^f^

Values are means ± standard deviation of three replicates. The different letters on the same column show a significant difference according to Duncan’s test at *p* ≤ 0.05.

## Data Availability

The original contributions presented in the study are included in the article, further inquiries can be directed to the corresponding author.

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
