# Peer review of "Assessment of the Nutritional Benefits and Aflatoxin B1 Adsorption Properties of Blackberry Seed Cold-Pressed Oil By-Product"

_foods, 2024, doi:10.3390/foods13193140_

Round 1

Reviewer 1 Report

Comments and Suggestions for Authors

Dear authors, here are my comments/questions:

this paper explores the potential valorization of blackberry seed oil cake,  as a nutritionally valuable 15 material with aflatoxin B1 (AFB1) adsorption properties.

Why do you study only one aflatoxin (AFB1) and not other too? Especially OTA which quite common?

Methods: in the dexcription of DPPH, ABTS, RP you mention you take 10μL of sample. But the sample is dry powder. Do you solubilize it in a solvent? 

TAble 1. please clarify a) the addition of components sums up to 94%, why not 100% and the ADF , ADL etc are % of what? of the total mass?

Line 293. Can someone actually consume 100g of the cake? It is a misleading suggestion

If you calculate IC50 of the antioxidant power isbt it a more standardized method?

Table 4 is mixed up with the numbering of the manuscript lines

Fianlly please improve the plagiarism % , which now is 39%.

Comments on the Quality of English Language

minor editing

Author Response

The authors would like to thank the reviewer for their professional and helpful remarks. The authors have accepted the reviewer’s suggestions and made revisions to the paper accordingly. We believe that these changes will meet the editor's and reviewers' criteria and that the revised paper will be interesting for new in the Foods.

Comments: Why do you study only one aflatoxin (AFB1) and not other too? Especially OTA which quite common?

Authors: The presented research offers a comprehensive overview, including not only an analysis of adsorption properties but also an insight into the nutritional benefits of blackberry seed oil cake (BBSOC). Previous studies conducted by the authors focused on the examination of aflatoxin B1, which has become increasingly prevalent in cereals in the region where the authors are based (doi/10.1002/jsfa.9601, doi/abs/10.3920/WMJ2017.2229). The AFB1, used in these experiments, is the crude toxin synthesized in the laboratory from a toxigenic isolate of Aspergillus flavus derived from wheat grains (10.1556/066.2020.49.4.3). Future research will expand the scope to include the adsorption of various mycotoxins such as ochratoxin A, deoxynivalenol, fumonisins… This will require a comprehensive examination of the sorption mechanisms, leading to a more detailed understanding of the adsorption processes.

Miljanić (Krulj), J., Markov, S., Bočarov‐Stančić, A., Pezo, L., Kojić, J., Ćurčić, N., Janić Hajnal, E., & Bodroža‐Solarov, M. (2019). The effect of storage temperature and water activity on aflatoxin B1 accumulation in hull‐less and hulled spelt grains. Journal of the Science of Food and Agriculture, 99(7), 3703-3710.

Miljanić (Krulj), J., Đisalov, J., Bočarov-Stančić, A., Pezo, L., Kojić, J., Vidaković, A., & Solarov, M. B. (2018). Occurrence of aflatoxin B1 in Triticum species inoculated with Aspergillus flavus. World Mycotoxin Journal, 11(2), 247-257.

Miljanić (Krulj), J., Ćurčıć, N., Stančıć, A. B., Kojıć, J., Pezo, L., Tukuljac, L. P., & Solarov, M. B. (2020). Molecular identification and characterisation of Aspergillus flavus isolates originating from Serbian wheat grains. Acta Alimentaria, 49(4), 382-389.

Methods: in the description of DPPH, ABTS, RP you mention you take 10μL of sample. But the sample is dry powder. Do you solubilize it in a solvent?

Authors: Thank you for this very important comment! The antioxidant assays were performed on 70% methanol extracts of BBSOC dry powder. We added this important information in manuscript, and we apologize for this omission.

Table 1. please clarify a) the addition of components sums up to 94%, why not 100% and the ADF , ADL etc are % of what? of the total mass

Authors: Yes, the results are expressed on the total mass. In order to present the units uniform in Table 1, % has been changed to g/100g.

Chemical composition analysis is done by methods that have a certain measurement uncertainty. The measurement uncertainty comes from the equipment, the purity of the chemicals, the analyst and other factors that introduce small errors into the result. That is the reason why the sum of the components is not exactly 100%. Most of the used methods are accredited by the national accreditation body and are checked every year.

Line 293. Can someone actually consume 100g of the cake? It is a misleading suggestion.

Authors: Corrected. The iron content in BBSOC flour was 123.48 mg/kg, indicating that consuming BBSOC can contribute to meeting part of the recommended daily intake of 14 mg.

If you calculate IC50 of the antioxidant power isbt it a more standardized method?

Authors: Thank you for this comment. We chose to express antioxidant activity as Trolox equivalents because Trolox is a widely recognized standard in the field. Its use allows for consistent and comparable measurement of antioxidant activity across different studies, as it provides a direct reference point relative to a known antioxidant.

Table 4 is mixed up with the numbering of the manuscript lines

Authors: Corrected, manuscript lines are assigned during publisher manuscript processing.

Finally please improve the plagiarism %, which now is 39%.

Authors: Corrected, the manuscript have been revised, repetition rate is reduced.

Reviewer 2 Report

Comments and Suggestions for Authors

Full text English needs to be improved, the mechanism of by-products of blackberry seed cold-pressed oil to absorb aflatoxin B1 needs to be in-depth analysis, and it is best to supplement the nuclear magnetic data.

Comments on the Quality of English Language

poor.

Author Response

The authors would like to thank the reviewer for their professional and helpful remarks. The authors have accepted the Reviewer’s suggestions and made revisions to the paper accordingly. We believe that these changes will meet the Editor's and Reviewers' criteria and that the revised paper will be interesting for new in the Foods.

The text of the manuscript is checked by a native English speaker as required according to comments. This includes detailed improvements in grammar and overall readability to enhance the comprehensibility of the manuscript.

In addition, we have addressed your suggestions and will implement the following improvements in our future work. We will conduct a comprehensive analysis of the by-products of blackberry seed cold-pressed oil to better understand their role in absorbing aflatoxin B1. This will include an in-depth examination of the mechanisms involved and will be supplemented with nuclear magnetic data to provide a more detailed insight. Research will broaden the scope to include the adsorption of various mycotoxins beyond aflatoxin B1.

Round 2

Reviewer 2 Report

Comments and Suggestions for Authors

The authors have corrected the paper carefully, I recommend the acceptance of the paper at present status.